# Malaria parasites differentially sense environmental elasticity during transmission

Johanna Ripp[1] (iD), Jessica Kehrer[1] (iD), Xanthoula Smyrnakou[1,2], Nathalie Tisch[3,4] (iD), Joana Tavares[5,6] (iD), Rogerio Amino[6], Carmen Ruiz de Almodovar[3,4] (iD) & Friedrich Frischknecht[1,*] (iD)

## Abstract

Transmission of malaria-causing parasites to and by the mosquito relies on active parasite migration and constitutes bottlenecks in the *Plasmodium* life cycle. Parasite adaption to the biochemically and physically different environments must hence be a key evolutionary driver for transmission efficiency. To probe how subtle but physiologically relevant changes in environmental elasticity impact parasite migration, we introduce 2D and 3D polyacrylamide gels to study ookinetes, the parasite forms emigrating from the mosquito blood meal and sporozoites, the forms transmitted to the vertebrate host. We show that ookinetes adapt their migratory path but not their speed to environmental elasticity and are motile for over 24 h on soft substrates. In contrast, sporozoites evolved more short-lived rapid gliding motility for rapidly crossing the skin. Strikingly, sporozoites are highly sensitive to substrate elasticity possibly to avoid adhesion to soft endothelial cells on their long way to the liver. Hence, the two migratory stages of *Plasmodium* evolved different strategies to overcome the physical challenges posed by the respective environments and barriers they encounter.

**Keywords** cell migration; ookinete; sporozoite; substrate elasticity
**Subject Category** Microbiology, Virology & Host Pathogen Interaction
See also: **A Vaughan** (April 2021)

## Introduction

*Plasmodium* parasites need to invade host cells and migrate across tissues at different stages throughout their life cycle (Fig EV1A). Their substrate-dependent form of locomotion, termed gliding motility, allows for very fast cell migration that does not involve any cellular protrusions and is propelled by an actomyosin motor (Heintzelman, 2015). The parasite forms developing inside the mosquito midgut after an infectious blood meal, the ookinetes, have to leave the blood meal and traverse the midgut epithelium to transform into oocysts at the basal membrane (Zieler & Dvorak, 2000; Vlachou *et al*, 2004; Kumar & Barillas-Mury, 2005; Kan *et al*, 2014). These ookinetes already move actively within the soft blood meal before migrating into the epithelium (Trisnadi & Barillas-Mury, 2020; Volohonsky *et al*, 2020). Inside the oocyst, sporozoites are formed which can infect vertebrates during a second blood meal of the mosquito. When fully developed, these sporozoites move inside the oocyst and egress (Klug & Frischknecht, 2017). They are transported within the mosquito circulatory system until they attach to and invade the salivary gland (Pimenta *et al*, 1994). There is only little movement inside the gland but once the sporozoites are transmitted into the skin, they start to migrate at high speed within the dermis (Frischknecht *et al*, 2004; Amino *et al*, 2006; Hopp *et al*, 2015). When a sporozoite finds a blood capillary, it can enter the blood stream, which transports the sporozoites to the liver (Frevert *et al*, 2005; Tavares *et al*, 2013). After exiting the circulation and traversal of several hepatocytes, the sporozoite finally invades a hepatocyte and develops into thousands of red blood cell infecting merozoites (Prudêncio *et al*, 2006; Tavares *et al*, 2013). How the motile *Plasmodium* stages recognize their target tissues and how the environment influences motility is largely unknown. Several sporozoite proteins have been found to be important for salivary gland invasion (Aly *et al*, 2009). Whether there is a specific receptor mediating tissue-specific invasion of salivary glands is currently under debate (O'Brochta *et al*, 2019; Klug *et al*, 2020). Liver invasion is thought to be triggered by highly sulfated heparan sulfate proteoglycans which protrude through fenestrations of endothelial cells lining the liver sinusoids (Pradel *et al*, 2002; Coppi *et al*, 2007; Tavares *et al*, 2013).

*In vivo* imaging has been used to characterize *Plasmodium* motility in its natural environment (Frischknecht *et al*, 2004; Amino *et al*, 2006; Hopp *et al*, 2015; Trisnadi & Barillas-Mury, 2020), but it does not allow the tuning of individual biochemical or mechanical properties. In contrast, various *in vitro* assays

1 Integrative Parasitology, Center for Infectious Diseases, Heidelberg University Medical School, Heidelberg, Germany
2 Gene Therapy for Hearing Impairment and Deafness, Department of Otolaryngology, Head & Neck Surgery, University of Tübingen Medical Center, Tübingen, Germany
3 Biochemistry Center, Heidelberg University, Heidelberg, Germany
4 European Center for Angioscience (ECAS), Medical Faculty Mannheim, Heidelberg University, Mannheim, Germany
5 IBMC-Institute for Molecular and Cell Biology, i3S - Institute for Research and Innovation in Health, University of Porto, Porto, Portugal
6 Malaria Infection and Immunity Unit, Department of Parasites and Insect Vectors, Institut Pasteur, Paris, France
*Corresponding author. Tel: +49 6221 566537; Fax: +49 6221 564643; E-mail: freddy.frischknecht@med.uni-heidelberg.de

allowed to examine the influence of defined environmental factors such as ligand density, hydrodynamic flow, substrate topography, and elasticity on sporozoite motility (Hegge *et al*, 2010; Hellmann *et al*, 2011; Perschmann *et al*, 2011; Muthinja *et al*, 2018). This revealed that sporozoites move best on stiff substrates such as glass or gels above 70 kPa with intermediate ligand density, i.e. ligands spaced between 50 and 100 nm apart (Perschmann *et al*, 2011). These studies also showed that sporozoites follow environmental topographical cues on their way through the skin (Hellmann *et al*, 2011) and slow motility under flow (Hegge *et al*, 2010). However, these studies were limited to 2D surfaces, did not investigate ookinetes, and failed to compare the small but possibly important elasticity differences relevant for cells encountered by sporozoites. Indeed, the elasticity of human tissues varies from soft brain tissue to intermediate dermal to stiff bone tissue (Engler *et al*, 2006; Zahouani *et al*, 2009) with endothelial cells lining the interior of blood vessels being softer than dermal fibroblasts (Grady *et al*, 2016) (Fig EV1B).

Here, we introduce tunable polyacrylamide (PA) hydrogels to investigate *Plasmodium* ookinete and sporozoite motility on and within elastic 2D and 3D substrates (Fig EV1C). As opposed to natural hydrogels such as Matrigel or collagen, PA hydrogels can be manufactured with defined elasticity and pore size by adjusting the monomer to crosslinker concentration (Holmes & Stellwagen, 1991; Tse & Engler, 2010; Wen *et al*, 2014). This allowed us to mimic different microenvironments and study the effect of single physiologically relevant physical parameters on motility of wild-type and mutant parasites. We show that ookinetes change their migration path to disseminate faster on stiffer substrates possibly reflecting a change from the soft blood meal to the stiffer midgut epithelium. Also, we find that ookinetes can actively migrate for over 1 day and that the much shorter lived sporozoite motility in 3D hydrogels mimics their migration in the skin. Importantly, we find that the elastic nature of endothelial cells limits the capacity of sporozoites to move on them, suggesting that sporozoites evolved to avoid adhesion to soft endothelial cells in order to ensure long-distance dispersion within the blood flow and efficient homing to the liver.

# Results

## Substrate stiffness impacts ookinete migration pattern

Ookinetes need to exit the midgut lumen and traverse the epithelium to establish oocysts *in vivo*. Their migratory behavior differs depending on whether they glide inside the lumen or whether they are in contact with the midgut surface (Trisnadi & Barillas-Mury, 2020). To investigate the effect of mechanical substrate properties on ookinete motility, we generated non-adhesive PA hydrogels with various elasticities ranging from approximately 1–40 kPa and performed time-lapse live cell imaging using a wide field fluorescence microscope for either fluorescence, phase- or differential interference contrast (DIC) imaging (Engler *et al*, 2006; Tse & Engler, 2010). We indeed observed reduced adhesion of ookinetes to these non-coated planar PA hydrogels and no continuous movement. We therefore sandwiched the parasites between two hydrogels to establish a confined environment (Fig 1A). This

confinement was sufficient to induce ookinete motility in the absence of specific ligands (Movie EV1). As opposed to experiments on glass, where only about half of the ookinetes moved, nearly all ookinetes were motile if sandwiched between two elastic PA hydrogels independent of gel elasticity (Fig 1B). The speed was enhanced 2- to 3-fold on elastic PA hydrogels compared to glass with only minor differences between soft and stiff hydrogels as analyzed by manual tracking (Fig 1C). Interestingly, ookinetes were motile for over 20 h on hydrogels, while they already stopped moving on glass within about 3 h (Fig 1D). After 20–26 h of motility, a higher fraction of ookinetes was still motile on soft than on stiff gels but they were fastest on stiff gels (Fig 1D and E). Strikingly, ookinetes were mostly moving in circular trajectories on soft hydrogels while they were moving more directional on stiffer substrates (Fig 1F). This resulted in a higher mean square displacement of ookinetes moving on stiffer hydrogels (Fig 1G). We hypothesize that this switch in behavior from circular to linear migration as well as from long to short-lived migration might play a role *in vivo* as ookinetes move out of the soft blood bolus and encounter a stiffer layer of epithelial cells. Adaption to substrate stiffness could also play a role once the ookinete has passed through the epithelial cells and faces the basal lamina (Han *et al*, 2000; Vlachou *et al*, 2004).

## 3D hydrogels support skin-like sporozoite migration

Upon transmission, sporozoites move at high speed through the dermis until they find and invade a blood capillary (Amino *et al*, 2006; Hopp *et al*, 2015). We probed if PA hydrogels could be used for the study of motile sporozoites in 3D. Large metazoan cells do not enter into synthetic hydrogels as they cannot degrade these gels or squeeze through the pores. In line with this, we did not observe any ookinetes penetrating the PA hydrogels. However, when infected salivary glands were sandwiched between a soft PA hydrogel and a glass coverslip, we observed sporozoites moving into the hydrogel (Fig 2A). Inside these hydrogels, they mainly moved in helical trajectories (Fig 2B, Movie EV2) similarly to what has been observed in Matrigel (Amino *et al*, 2008). As sporozoites move very fast, image acquisition to follow their paths is challenging. Hence, sporozoites that moved horizontally to the plane of image acquisition could readily be followed, while those that moved vertically could not be tracked over time as they moved quickly out of focus; some sporozoites also stopped moving. To study the effect of pore size on sporozoite motility, we manufactured soft hydrogels with different concentrations of crosslinker, thereby tuning the pore size (Tse & Engler, 2010; Wen *et al*, 2014). With decreasing pore size, sporozoites slowed down and a higher fraction of sporozoites became immotile. Eventually, at small pore sizes, sporozoites failed to enter the hydrogels. Sporozoite motility is induced by bovine serum albumin (BSA) (Vanderberg, 1974). Adding BSA to the medium was essential for sporozoites to move inside the hydrogels, but BSA was not necessary for sporozoites to enter hydrogels. This suggests that substrate contact and/or substances released from the salivary gland are sufficient for surface motility. Importantly, sporozoite speed in 3% AA/0.06% BIS (small pore) or 3% AA/0.03% BIS (large pore) hydrogels was similar to the speed distribution observed in the skin *in vivo*, with the sporozoites in the small pore gel representing the slower

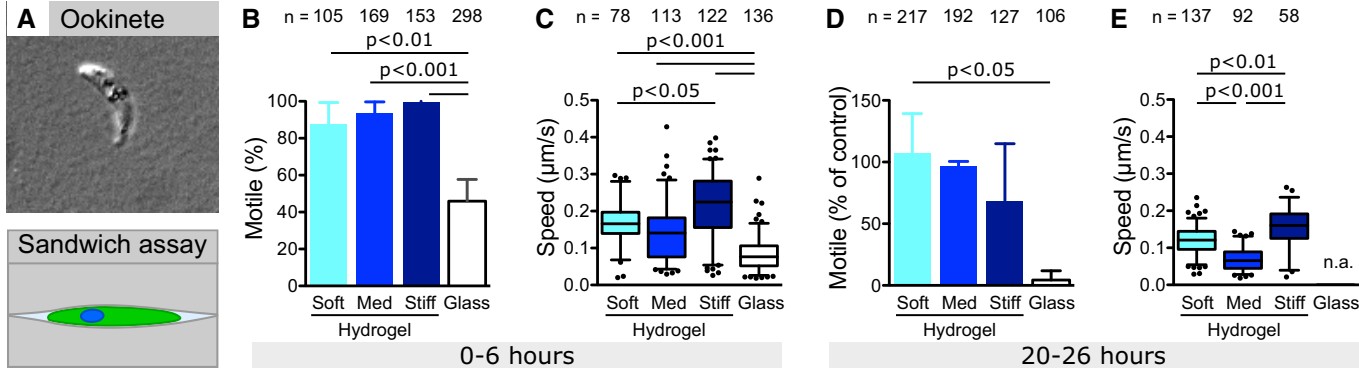

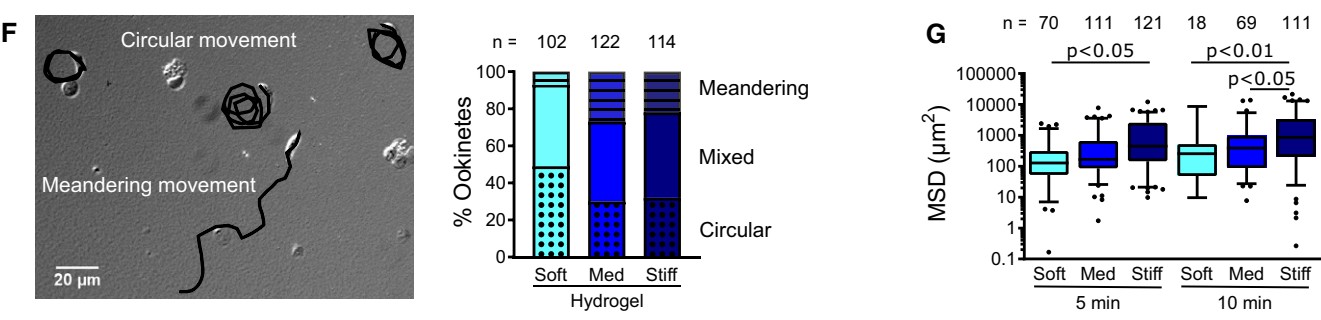

**Figure 1. Substrate stiffness affects motility patterns of ookinetes.**

A  Ookinetes sandwiched between two hydrogel surfaces.

B  Percentage of ookinetes moving for more than one parasite length within the time of observation (5–10 min) if sandwiched between soft, medium (med) or stiff hydrogels or glass. Bars represent the mean and error bars the standard deviation of at least three independent experiments.

C  Speed of motile ookinetes on substrates of different stiffnesses as indicated.

D  Percentage of motile ookinetes after 20–26 h of incubation between hydrogels compared to parasites directly imaged after setting up the experiment as shown in (B). Shown is the average ± SD. from at least two independent experiments.

E  Speed of motile ookinetes after 20–26 h of incubation between hydrogels. Speed on glass was not analyzed (n.a.) due to the low fraction of motile ookinetes at this timepoint.

F  Migration patterns of ookinetes. Image: Overlay of tracked migration paths (black) and DIC image showing ookinetes at the end of the recorded image sequence. Graph: Quantitative analysis of the indicated different migration patterns on soft, medium (med) and stiff hydrogels.

G  Mean square displacement of ookinetes moving on hydrogels of different stiffnesses at two timepoints.

Data information: Numbers above indicate the number of ookinetes analyzed. Significance for (B) and (D) determined by one-way analysis of variance with Bonferroni's multiple comparison test. Significance for (C), (E) and (G) determined by Kruskal–Wallis test with Dunn's multiple comparison test. Box-and-whisker plots (C, E, G) depict the 25% quantile, median, 75% quantile, and nearest observations within 1.5 times the interquartile range (whiskers). Outliers beyond this range are shown as black dots. Source data are available online for this figure.

*in vivo* fraction, while the sporozoites in the large pore gel were representing the faster fraction (Fig 2C). To test how accurately soft PA hydrogels can be used to mimic sporozoite migration in the skin, we analyzed two transgenic parasite lines lacking the actin-binding protein coronin (Bane *et al*, 2016) and the heat shock protein 20 (HSP20) (Montagna *et al*, 2012), respectively. We chose these lines as they show different motility phenotypes in the skin and on glass: *coronin(-)* sporozoites move well in the skin but show a migration defect on glass (Bane *et al*, 2016). In contrast, *hsp20(-)* sporozoites move slower both on glass and within the skin (Montagna *et al*, 2012). Interestingly, *coronin(-)* sporozoites moved as well as wild-type sporozoites in PA hydrogels with small pores but significantly slower in the PA hydrogel with large pores (Fig 2D and E). This might reflect the decreased capacity of *coronin(-)* sporozoites to attach to surfaces (34), which could lead to impaired gliding in gels with large pores (i.e. less available

substrate surface), but is compensated by gels with small pores (i.e. more available substrate surface). In contrast, *hsp20(-)* sporozoites moved inefficiently in both gels (Fig 2D and E). This shows that sporozoite migration parameters in the PA hydrogels largely mimicked those observed in the skin in all three investigated parasite lines. To further test the versatility of the gel system, we assessed if drugs can diffuse into the hydrogel and affect motility. Indeed, the actin polymerization inhibitor cytochalasin D (CytoD), which inhibits migration *in vitro* (Münter *et al*, 2009), also reduced the speed of sporozoites moving in PA hydrogels at a low concentration of 25 nM (Fig 2F and G), while at 50 nM sporozoites did not enter the gels. Together, these data suggest that the 3D PA hydrogels can be used to mimic key features of sporozoite migration in the skin and can serve as model system to rapidly test drugs or antibodies against the parasite during the first minutes of a *Plasmodium* infection.

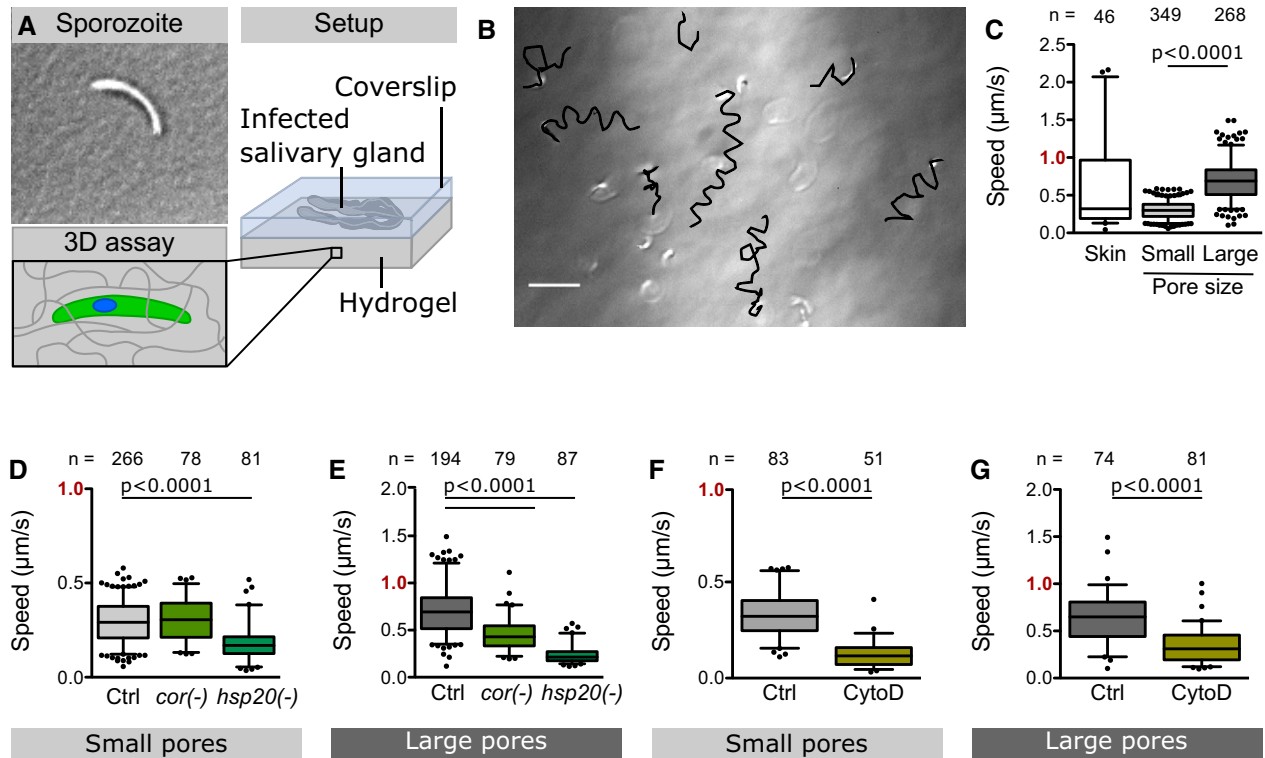

**Figure 2. Sporozoite motility inside 3D hydrogels with different pore sizes.**

A   Infected salivary glands were sandwiched between a soft PA hydrogel and a glass coverslip inducing sporozoites to move into the hydrogel.
B   Overlay of sporozoite tracks (black) and DIC image showing sporozoites inside a soft hydrogel at the end of a recorded image sequence. Scale bar, 20 μm.
C   Speed of WT sporozoites moving in the skin *in vivo* (Douglas *et al*, 2018b) and in soft hydrogels with small or large pores.
D, E   Speed of indicated mutant and control (Ctrl) parasites in hydrogels with small (D) and large (E) pores.
F, G   Speed of drug-treated and control parasites in hydrogels with small (F) and large (G) pores.

Data information: Note the faster speed in gels with larger pores. Data from at least two independent experiments. Numbers above bars indicate the number of sporozoites analyzed. Significance determined by Mann–Whitney test. Box-and-whisker plots (C–G) depict the 25% quantile, median, 75% quantile, and nearest observations within 1.5 times the interquartile range (whiskers). Outliers beyond this range are shown as black dots.

Source data are available online for this figure.

## Diminished sporozoite motility on endothelial cells and soft substrates

Upon entry into the blood circulation of a vertebrate host, the sporozoite needs to travel to the liver without "losing time" or "getting lost" by adhering to and migrating on the endothelial cells of the blood vessels or by crossing the endothelial cell layer and penetrating the surrounding tissue. How the sporozoite, which can migrate on virtually all biological surfaces, avoids to leave the blood stream before it reaches the liver is unclear. One possibility might be the lack of a receptor on endothelial cells that specifically allows adhesion under flow. Alternatively, the relative softness of endothelial cells (Grady *et al*, 2016) might inhibit adhesion and/or motility. We first attempted to study sporozoite motility on confluent human umbilical cord (HUVEC) endothelial cells, yet repeatedly failed to see any movement in different media. We next tried confluent layers of cultured endothelial cells from mouse brain (bEnd.3) and human skin (HDMEC) as well as pericytes, a type of mural cell that wraps around endothelial cells lining capillary blood vessels. Again no robust sporozoite migration was observed raising the question whether

sporozoite indeed cannot move on vascular cells. Next, we imaged sporozoite on semi-confluent liver-derived endothelial cells (LSECs). This showed some sporozoites on the cells and others on the glass surface next to the cells and revealed that those attached to the cells were moving much less than those on glass (Fig 3A). We then cultivated confluent layers of human foreskin fibroblasts (HFF), and HUVEC cells and quantified the motion of sporozoites derived from the same mosquitoes (Fig 3B). Again, this showed literally no sporozoite movement on the HUVEC cells, while sporozoites moved fine on HFF cells or on glass (Fig 3C). Hence, these data suggest that sporozoites lack the capacity to migrate efficiently on cells forming blood vessels. To probe if this decrease of motility could be due to substrate stiffness, we tested whether a physiologically relevant range of elasticities affects sporozoite motility. To this end, we performed cell migration assays on soft, intermediate, and stiff 2D hydrogels (Fig 3D and E), with the soft hydrogels (5% AA/0.03% BIS) reflecting closest the elasticity of endothelial cells and the intermediate hydrogels (5% AA/0.3% BIS) reflecting the elasticity of dermis and liver (Engler *et al*, 2006; Tse & Engler, 2010; Grady *et al*, 2016). Interestingly, the motile fraction of sporozoites significantly

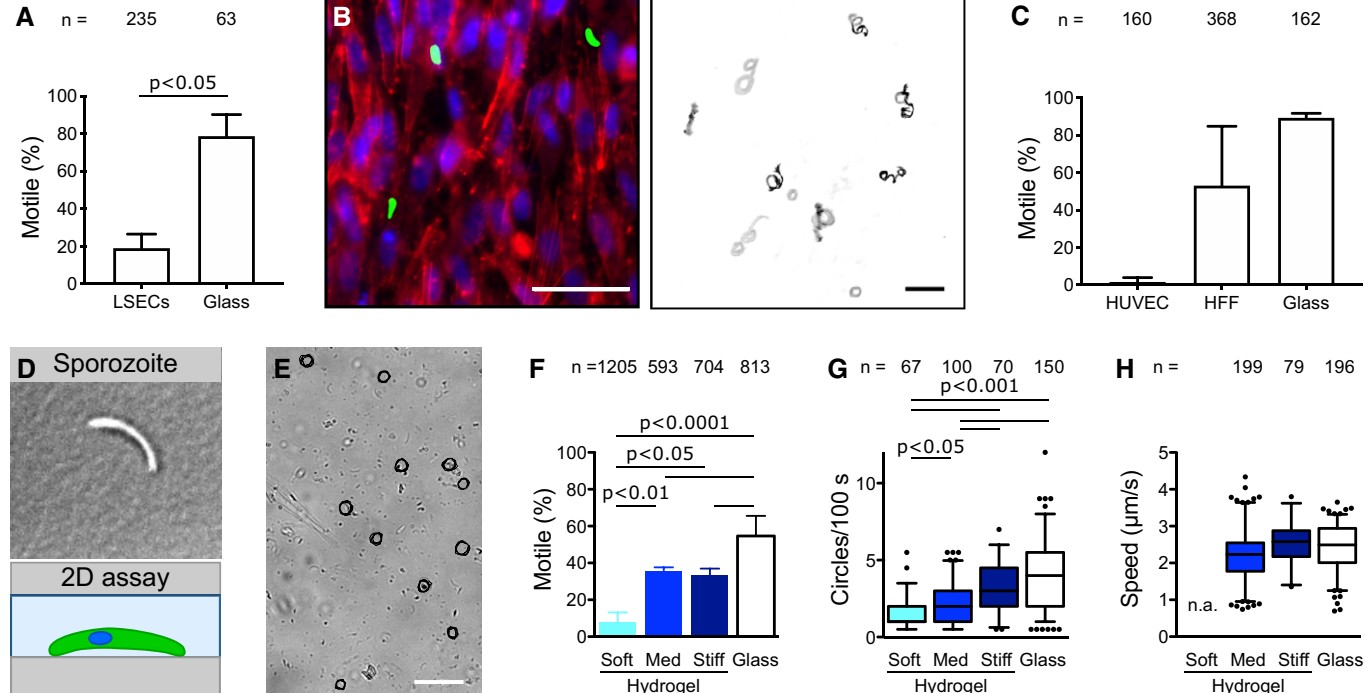

**Figure 3. Sporozoite motility on cells and 2D hydrogels of different stiffnesses.**

A   Percentage of motile sporozoites on endothelial cells (LSECs) versus glass. Shown is average ± SD. from three independent experiments. Significance determined using an unpaired t-test.

B   Left: Merged fluorescence images showing sporozoites on cultured HFF cells. Actin filaments of HFF cells were stained with phalloidin (shown in red), nuclei were stained with Hoechst (blue) and sporozoites expressed GFP (green). Scale bar, 50 μm. Right: Maximum projection of sporozoites moving for 3 min on cultured HFF cells. Scale bar, 50 μm.

C   Percentage of motile sporozoites on confluent layers of endothelial cells (HUVEC), fibroblasts (HFF) and on glass. Shown is average ± SD. from two independent experiments. No significant differences as determined by one-way analysis of variance with Bonferroni's multiple comparison test.

D   Sporozoite on planar uncoated hydrogels.

E   Overlay of sporozoite tracks (black) and DIC image showing sporozoites on a hydrogel. Scale bar, 50 μm.

F   Percentage of motile sporozoites on soft, medium (med) or stiff hydrogels or glass 30 min after activation with BSA. Shown is the average ± SD. from at least three independent experiments. Significance determined by one-way analysis of variance with Bonferroni's multiple comparison test.

G   Migration persistance as determined by the number of circles performed within 100 s on hydrogels of different stiffnesses and glass 30 min after activation with BSA. All sporozoites moving for more than one parasite length within the time of observation were analyzed, even if they stopped moving or detached during imaging. Significance determined by Kruskal–Wallis test with Dunn's multiple comparison test.

H   Speed of sporozoites moving consistently for at least 60 s on substrates of different stiffnesses. Note that not all sporozoites analyzed in (G) complied with that requirement. To be able to analyze a high number of sporozoites, movies were taken 10–45 min after activation with BSA. Speed of sporozoites on soft hydrogels was not analyzed (n.a.) due to the low fraction of motile sporozoites on these hydrogels. No significant differences as determined by Kruskal–Wallis test with Dunn's multiple comparison test.

Data information: Numbers above bars indicate the number of sporozoites analyzed. Box-and-whisker plots (G, H) depict the 25% quantile, median, 75% quantile, and nearest observations within 1.5 times the interquartile range (whiskers). Outliers beyond this range are shown as black dots.
Source data are available online for this figure.

dropped from medium to soft hydrogels but remained constant between medium and stiff hydrogels (Fig 3F). The number of circles traced by migrating sporozoites, a combined measure for persistent migration and speed, increased with increasing substrate stiffness (Fig 3G). As substrate stiffness did not affect the speed of continuously moving sporozoites (Fig 3H), the decreased stiffness diminishes persistence of movement. Together, these data suggest that sporozoites are less capable to move for long periods of time on endothelial cells, which could partly be due to the lower stiffness of these cells. This apparent lack of migration capacity on endothelial cells could provide a key evolutionary advantage in allowing efficient transport of sporozoites to the liver, which is essential for efficient propagation of the life cycle.

# Discussion

### Ookinetes but not sporozoites display different modes of motility dependent on substrate elasticity

Here, we show that ookinetes and sporozoites are able to move at fast speeds if confined between or within uncoated PA hydrogels. Tuning of these PA gels to elasticities as they occur on the natural substrates encountered by the parasites revealed intriguing differences between the motile behavior of ookinetes and sporozoites. While ookinetes moved at relatively slow speed for over 20 h on soft gels, sporozoites moved at ten-fold higher speed for < 1 h. Furthermore, ookinetes adapted their migration path to substrate

stiffness, while sporozoites either migrated on their typical circular paths or did not migrate at all. Adaptations of migration paths in ookinetes might be an adaptation to the very different requirements the ookinetes have to master in their short life. Firstly, they need to exit the blood meal, a process that appears to depend on gravity in mosquitoes that agglutinate their blood meal. This was revealed by placing mosquitoes on their heads after a blood meal, leading to most oocysts forming at the anterior midgut. In contrast, in mosquitoes that do not agglutinate their blood oocyst formation is independent of gravity and distributed evenly along the midgut (Shute, 1949; Cociancich et al, 1999; Kan et al, 2014). These experiments suggest that active migration through the blood meal might well play a role in ookinete dissemination as diffusion is likely inefficient. Our results showing that ookinetes migrate equally well on the softest gel as they do on the hardest substrate (glass), suggest that indeed their motility apparatus and shape might have evolved to allow active dissemination within the blood bolus. In vivo, this might not manifest itself as persistent migration but might be limited to the occasional displacement of a red blood cell or the agglutinated debris, with the parasite progressing slowly across the lumen of the midgut toward the epithelial cells at the edge. Indeed, active ookinete motility has recently been demonstrated within the blood meal by in vivo imaging (Trisnadi & Barillas-Mury, 2020). However, more long-term imaging needs to be performed to understand how this movement leads to ookinete displacement in the blood meal.

The observation that ookinetes move in a predominantly circular manner on the softest gels yet in a predominantly linear fashion on harder surfaces suggests that the parasite is sensing the stiffness of its environment and adapts its motile behavior accordingly. This could be relevant as the parasite migrates from the soft blood meal through the less soft epithelium or as it seeks to arrest below the basal lamina, which is likely the least elastic substrate the ookinete encounters. To understand the physiological relevance of this switch in migration path, it would now be interesting to screen transgenic ookinetes with documented defects in migration for their capacity to undergo this transition from circular to linear motion and to correlate this with in vivo observations.

## A synthetic 3D environment to mimic sporozoite migration in the skin

We found that ookinetes do not move on planar uncoated PA hydrogels, while sporozoites can move continuously on stiff PA hydrogels. Indeed, most laboratories use a Matrigel assay to investigate ookinete migration suggesting that ookinetes need a 3D environment to migrate robustly (Kan et al, 2014; Brochet et al, 2014). In contrast, sporozoite motility has been intensively characterized on just a 2D glass surface since its discovery in 1964 (Yoeli, 1964). This indicates that there is an important difference in parasite motility dependent on the dimensionality of the environment, which might be due to a different capacity to build adhesion sites or due to the arrangement of organelles. In contrast to ookinetes, sporozoites show a very clear chiral cellular architecture with strongly bent polar rings, which we suggested to play a role in robust 2D motility (Kudryashev et al, 2012; Kan et al, 2014). Our data imply that adhesive interactions are especially important for motility on 2D substrates, while 3D motility of the parasites appears irrespective of receptor–ligand interactions. Unlike ookinetes, sporozoites do get in

contact with planar substrates (transition from hemolymph to salivary gland and blood circulation to liver) as well as 3D environments (skin) on their journey from mosquito to mammal.

Interestingly, we found sporozoites moving within soft PA hydrogels, which feature narrow pores (Wen et al, 2014), indicating that substrate softness does not impair motility in a 3D setting. Sporozoites slowed down in hydrogels with decreasing pore size, likely due to steric hindrance imposed by the polymeric network. However, we cannot exclude that sporozoites also sense and respond to the substrate and modulate their behavior. This could be addressed by investigating sporozoites with, e.g., calcium-sensitive dyes as they migrate through the gels, as previous work showed that calcium is elevated in sporozoites migrating on flat surfaces in vitro (Carey et al, 2014). Also, motility clearly depends on a rapid turnover (i.e. formation and disruption) of adhesion sites (Münter et al, 2009). Sporozoites forming very tight adhesions are slower than those forming weaker ones, while those forming very weak adhesions cannot move. Hence, sporozoites with a slight defect in adhesion might not move on a flat and stiff substrate anymore as seen for coronin(-) sporozoites (Bane et al, 2016) but are still moving fine in a 3D environment.

Sporozoite speed inside these hydrogels was similar to the range of speeds observed in vivo in the skin (Amino et al, 2006; Hopp et al, 2015; Douglas et al, 2018b) or natural hydrogels (Amino et al, 2008). Yet, their trajectories were more regular than in the skin, which might be explained by the heterogenous composition of the skin consisting of different extracellular matrix proteins, fibers, and cell types. The robust migration within the gels shows that BSA is sufficient to activate 3D motility of sporozoites and, importantly, that there is no need for specific receptors. While such specific interactions between substrate proteins and sporozoite surface proteins clearly exist, they might only play minor modulating roles during migration as shown for TRAP (Dundas et al, 2018). Indeed, leukocytes are able to move in the absence of integrins through confined environments, but not on planar substrates, suggesting that receptor-ligand interactions are less important for 3D motility of cells in general (Lämmermann et al, 2008; Reversat et al, 2020).

To probe how well PA hydrogels can reconstitute sporozoite migration in the skin, we used two mutant parasite lines that show different in vitro and in vivo migration capacities. Interestingly, we found that confinement can fully compensate for the lack of the actin-binding protein coronin as was previously shown by in vivo imaging in the skin (Bane et al, 2016). We also observed similar speed ranges of wild-type sporozoites and their susceptibility to motility inhibiting drugs. This suggests that the 3D PA hydrogels represent good platforms to investigate Plasmodium sporozoite migration ex vivo and might be useful for rapid drug and antibody testing.

## Sporozoites only move on stiffer substrates—a mechanism for avoiding premature adhesion?

Like ookinetes, sporozoites can also migrate on a range of different substrates. Motility on soft PA gels was first shown in a study that measured traction forces of migrating sporozoites (Münter et al, 2009) and raised the question how parasites link to their substrate. Several observations such as transient adhesions (Münter et al, 2009; Song et al, 2012), active processing of adhesive proteins

(Ejigiri *et al*, 2012), and sensitivity to lateral flow (Hegge *et al*, 2010) suggest that sporozoites avoid strong attachment to their respective substrates during migration in order to achieve their high speed of more than 1 μm/s (Vanderberg, 1974; Frevert *et al*, 2005; Amino *et al*, 2006). However, sporozoites need to attach specifically to target organs such as the salivary gland or the liver. Clearly, a complex machinery must regulate this switch in behavior, likely involving calcium and cGMP signaling, adhesins as well as cytoskeletal arrangements (Coppi *et al*, 2007; Hegge *et al*, 2012; Song *et al*, 2012; Carey *et al*, 2014; Govindasamy *et al*, 2016). This switch from migration to adhesion and invasion does not have to be abrupt but could also be gradual. It is currently not known if sporozoites pass by the salivary glands several times and how frequently they attach to and detach from the endothelium in the liver sinusoids before they finally attach and enter the organs. Nano-patterning of ligands on surfaces that do not allow adhesion shows that only a few dozen adhesins are necessary to allow sporozoite motility (Perschmann *et al*, 2011; Hellmann *et al*, 2013). This could suggest that the major surface protein CSP mediates low affinity unspecific binding to a surface with adhesins of the TRAP family providing for stronger but transient substrate adhesion (Münter *et al*, 2009; Ejigiri *et al*, 2012; Song *et al*, 2012; Frischknecht & Matuschewski, 2017; Klug *et al*, 2020). Clearly, modulation of adhesion and deadhesion cycles plays a complex role in motility and invasion. For example, *P. berghei* sporozoites lacking the actin-binding protein coronin rarely attached to and hence did not glide persistently on a flat substrate (Bane *et al*, 2016). This defect translated into a lower number of sporozoites entering the salivary gland but had no impact on sporozoite migration in the skin or liver invasion (Bane *et al*, 2016). This, and other evidence (Moreau *et al*, 2017; Douglas *et al*, 2018a), suggests that the salivary gland is a stronger barrier to sporozoite invasion than the liver.

The many tissues the sporozoite needs to traverse likely provide different barriers that necessitated a complex adaptation of the parasite. To be efficient, *Plasmodium* needed to evolve a migration machine that is superbly adapted. One of the ways sporozoites adapted appears the development of a cell surface that avoids strong adhesion. Two driving forces might play their part: First, a less-sticky surface might not be targeted by the complement system or antibodies. Second, the parasites avoid getting stuck before they reach their destination. Sporozoites are injected into the skin of a new host and enter the blood stream often at the periphery, e.g. at smelly, mosquito-attracting feet (Braks *et al*, 1999). While they need to migrate to cross the dermis and to enter the liver, they should not leave the blood stream on the long route from the bite site to the liver sinusoids. A mix of hydrodynamic flow, absence of specific ligands, and, as shown here, low adhesion to endothelial cells might combine to enhance the efficiency of homing to the liver (Fig EV2). Similarly, ookinetes could easily be trapped in the blood meal if they would attach too strongly to red cells and their digested remnants.

Recently, it was shown that membrane tension of red blood cells limits the entry of *Plasmodium* merozoites. A blood group variant that leads to higher membrane tension hence provides protection from severe malaria (Kariuki *et al*, 2020). The median tension of the Dantu red cells was at 800 pN/m about twice as high as the tension threshold allowing for invasion. Strikingly, fibroblasts are reported to be about twice as stiff as endothelial cells (Grady *et al*, 2016) suggesting that similar ranges of physical

parameters can impact *Plasmodium* parasites at different points in their life cycle. This provides a strong motivation to investigate the physical parameters of the parasites and their environment as drivers for adaptive evolution.

In conclusion, we show here that *Plasmodium* ookinetes migrate in different patterns on substrates of diverse stiffness and that sporozoites can migrate in PA hydrogels in ways mimicking their migration in the skin. This provides new *in vitro* assays to test early intervention strategies against malaria. We suggest that *Plasmodium* parasites evolved to finely tune their motility and adhesion machinery to avoid premature stickiness, which is important for efficient transmission to and from the mosquito vector of malaria.

# Materials and Methods

### Ethics statement

Animal experiments were performed according to FELASA and GV-SOLAS guidelines and approved by the responsible German authorities (Regierungspräsidium Karlsruhe).

### Parasite culture, mosquito infection, and parasite isolation

*P. berghei* parasite strains NK65 expressing GFP under the CS promoter, ANKA wt, *coronin(-)*, and *hsp20(-)* were used (Natarajan *et al*, 2001; Montagna *et al*, 2012; Bane *et al*, 2016). Ookinetes were cultured and purified as described before (Douglas *et al*, 2018a). *Anopheles stephensi* mosquitoes, which are the Indian mosquito species usually used in laboratory experiments with *P. berghei*, were infected and sporozoites isolated from salivary glands as described previously (Klug & Frischknecht, 2017). Experiments with salivary gland sporozoites were performed 17–25 days post-mosquito infection.

### Endothelial cell and pericyte cultures

Human umbilical vein endothelial cells (HUVECs; PromoCell), murine brain endothelial cells, bEND3, and human dermal microvascular cells, HDMEC, were cultured in Endopan 3 Kit for endothelial cells (PAN-Biotech) supplemented with 10% FBS, 100 U/ml penicillin, and 100 μg/ml streptomycin (both Gibco. by Life Technologies) in a 5% $CO_2$ humidified incubator at 37°C. Cells from passages 2 to 5 were used for the experiments. When indicated, cells were starved in growth factor-free Endopan 3 supplemented with 2% FBS, 100 U/ml penicillin, and 100 μg/ml streptomycin.

Human pericytes (kindly gifted by P. Carmeliet) were cultured in alphaMEM glutamax, 100 U/ml penicillin and 100 μg/ml streptomycin, 10% FBS, and 5 ng/ml PDFGβ (R&D) and used until passage 7.

### Preparation of uncoated polyacrylamide hydrogels

PA hydrogels were prepared as described before (Pelham & Wang, 1997). To tune the stiffness of the hydrogels (Tse & Engler, 2010), different monomer and crosslinker concentrations were added to the prepolymer solution. Soft gels (expected elastic modulus of approximately 1 kPa) were prepared with 5% AA/0.03% BIS, medium gels (about 9 kPa) with 5% AA/0.3% BIS, and stiff gels (about 40 kPa) with 8% AA/0.48% BIS (Tse & Engler, 2010). To increase the pore

**The paper explained**

**Problem**
Malaria parasites face different biochemical and physical environments in their complex life cycle forcing them to adapt to both parameters. Two forms of the parasite are highly motile yet one, the sporozoite, is much faster than the other, the ookinete. The ookinete needs to move out of the blood meal in a mosquito stomach, while the sporozoite is transmitted by the mosquito and needs to home to the liver, where it differentiates within hepatocytes. How does the sporozoite avoid attaching to a blood vessel outside the liver?

**Results**
Ookinetes are shown to migrate for many hours in mainly circular manner on very soft substrates mimicking the blood meal and to move on linear trajectories on harder substrates. Sporozoites are shown to migrate in 3D in soft substrates mimicking the skin, but fail to adhere to very soft substrates and a range of very soft vascular cells. This avoidance likely helps sporozoites for homing to the liver.

**Impact**
The presented assays will help to screen for transmission blocking antibodies to stop movement of either motile parasite stage.

size of soft gels, a gel formulation of 3% AA/0.06% BIS (small pores) or 3% AA/0.03% BIS (large pores) was used.

**Cell migration assays**

A small volume of medium containing ookinetes was sandwiched between two hydrogels. After an incubation time of 5–10 min, medium was removed from the side to confine the ookinetes between the hydrogels. For extended imaging periods, the sandwich was sealed using paraffin wax. Alternatively, sporozoites were pipetted into a silicone chamber placed on top of a hydrogel. For 3D hydrogel assays, whole infected salivary glands were dissected into 30 µl of medium on a glass coverslip ($22 \times 22$ mm) placed on top of a microscope slide. Subsequently, the salivary glands were covered with a hydrogel. As control, ookinetes and sporozoites were imaged on glass as described previously (Douglas *et al*, 2018a). As sporozoite motility decreases with increasing time of incubation in BSA containing activation medium, but the sporozoites also need some time to settle on the substrate and we could not centrifuge them down when doing experiments with hydrogels, we imaged sporozoites after 30 min of incubation on the different substrates at low magnification. For imaging on cells, $5 \times 10^4$ cells were seeded into 8-well Labtek chambered cover glass dishes. When cells reached at least 95% confluency, the RPMI or Endopan medium was replaced by medium containing 3% BSA and $3–5 \times 10^4$ freshly isolated sporozoites. Prior imaging dishes were centrifuged at 800 rpm for 5 min to allow sporozoites to adhere. Imaging was performed on an inverted Zeiss Axiovert 200 M microscope. Images of ookinetes were recorded every 20 s for 5–10 min, and sporozoites were imaged every 3 s for 2–3 min.

**Analysis**

Parasites were classified as motile if they moved for more than one parasite length during the time of observation. The percentage of motile sporozoites was determined by counting the total number of sporozoites within the first frame and dividing it by the number of parasites that moved at least half a circle as seen within the maximum projection of the image sequence. The speed was determined using the Manual Tracking plugin of Fiji (Schindelin *et al*, 2012). Only those parasites moving continuously for at least 60 s in the case of sporozoites or 5 min in the case of ookinetes were tracked. Mean square displacements of ookinetes were calculated from the position of the ookinetes in the first frame and in the following frames as given in the result files of the Manual Tracking plugin. Statistical analysis was performed in GraphPad Prism. Figures were generated using Inkscape.

# Data availability

The study includes no data deposited in external repositories.

**Expanded View** for this article is available online.

## Acknowledgements

We thank Miriam Reinig and the CEPIA of Institut Pasteur for production of *Anopheles stephensi* mosquitoes, Markus Ganter and Ulrich Schwarz for discussion and critically reading the manuscript. We would like to acknowledge the microscopy support from the Infectious Diseases Imaging Platform (IDIP) at the Center for Integrative Infectious Disease Research, Heidelberg, Germany. The work was funded by grants from the Human Frontier Science Program RGY0066/2015 (http://www.hfsp.org) to FF, the FRONTIER program of Heidelberg University (https://www.uni-heidelberg.de/de/forschung/forschungsservice/innovationsfonds-frontier) to FF and CRA, the Deutsche Forschungsgemeinschaft (DFG, German Research Foundation)—Projektnummer 240245660 - SFB 1129 (http://www.sfb1129.de) to FF, an ANR-DFG grant FR2140/11-1 to FF and RA, the French National Research Agency (grant no. ANR-10-JCJC-1302-PlasmoPEP), and the French Government's Investissement d'Avenir program, Laboratoire d'Excellence "Integrative Biology of Emerging Infectious Diseases" (grant no. ANR-10-LABX-62-IBEID) to RA. Individual funding from the Portuguese Foundation for Science and Technology through CEECIND/02362/2017 (to JT). FF and CRA are members of the CellNetworks Cluster of Excellence at Heidelberg University. JR and NT were members of the Heidelberg International Graduate School for the Biosciences (HBIGS), and XS was a member of the Master Program Molecular Biotechnology at Heidelberg University. Open Access funding enabled and organized by Projekt DEAL.

## Author contributions

JR, CRA, and FF designed the study. JR developed the hydrogels. JR and XS performed the hydrogel-based experiments. JK, NT, and JT performed the vascular cell-based experiments. FF, RA, and CRA coordinated the study. All authors analyzed and interpreted data related to this work. JR and JT performed the statistical analyses. JR and FF wrote the draft of the manuscript. All authors contributed to finalizing the paper.

## Conflict of interest

The authors declare that they have no conflict of interest.

## For more information

Author website: www.sporozoite.org.

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
