## [Review Process File · EMBO Molecular Medicine]

Malaria parasites differentially sense environmental elasticity during transmission

Johanna Ripp, Jessica Kehrer, Xanthoula Smyrnakou, Nathalie Tisch, Joana Tavares, Rogerio Amino, Carmen Ruiz de Almodóvar, and Friedrich Frischknecht

DOI: [10.15252/emmm.202113933](https://doi.org/10.15252/emmm.202113933)

Corresponding author: Friedrich Frischknecht (freddy.frischknecht@med.uni-heidelberg.de)

Review Timeline:	Transfer from Review Commons	11th Jan 21
	Editorial Decision:	15th Jan 21
	Revision Received:	28th Jan 21
	Accepted:	29th Jan 21

Editor: Zeljko Durdevic

Transaction Report:

This manuscript was transferred to EMBO Molecular Medicine following peer review at Review Commons.

Reviewer Reports (transfer from Review Commons)

Reviewer #1 (Evidence, reproducibility and clarity (Required)):

The authors establish a new strategy for assessing the motility of *Plasmodium berghei* ookinetes and sporozoites using 2D and 3D polyacrylamide hydrogels that mimic the environments encountered by each parasite stage in vivo. In vivo imaging experiments are difficult and do not easily allow for changes to experimental parameters, so this in vitro technique is important for characterizing the motility of these parasite stages. For example, the authors characterize the coronin(-) and hsp20(-) parasite lines, whose movement have previously been described in skin and on glass; the hydrogels mimic the elasticity of the skin environment, and so the motility of sporozoites in the hydrogels is a better model for motility than on glass slides. This in vitro strategy that mimics the in vivo environment may be useful in characterizing additional mutant parasite lines or antimalarial drugs that may cause defects in motility at either stage.

In addition to this methodology, the authors describe the motility of ookinetes and sporozoites and find that each stage uses different strategies for their motility, reflecting the different environments they encounter. The migration path of ookinetes is circular on soft surfaces and linear on stiffer surfaces, reflecting the movement of the ookinete from the soft bloodmeal in the midgut to the less soft epithelial cells of the midgut membrane. Sporozoites are motile for a short period of time, and do not migrate well on soft hydrogel or endothelial cells. The authors suggest the sporozoites are not adhering to the soft endothelial cells of the vasculature on their way to the liver (which is more stiff), and possibly plays a role in the sporozoites finding the correct tissue to invade.

****Major comments:****

Are the key conclusions convincing?

Their results are convincing and the authors describe the data appropriately. The number of cells analyzed for each experiment is adequate and experiments are appropriately controlled.

Should the authors qualify some of their claims as preliminary or speculative, or remove them altogether?

The authors appropriately describe their data, and use suitable language when speculating how the results of the hydrogel models relate to biological events in the life cycle of the motile parasite stages. Claims are appropriately tempered given the data presented.

Would additional experiments be essential to support the claims of the paper?

No additional experiments are needed to support the claims made in this work.

Are the data and the methods presented in such a way that they can be reproduced?

Yes, the materials and methods section describes how experiments were performed, or refers to the appropriate primary literature (Refs 25-28) to describe how data was generated and analyzed. Additional details here or in the main text that directly link an acrylamide/bisacrylamide formulation to specific cell/tissue types would clarify the use of each.

Are the experiments adequately replicated and statistical analysis adequate?

Yes, the authors have a large number/adequate number of cells counted for each figure, and use appropriate statistical tests to comment on significance.

Minor comments:

Are prior studies referenced appropriately?

In large part, prior publications are appropriately cited. Some additional citations would be helpful:

1-There could be a citation on line 65, about the in vivo imaging to characterize Plasmodium motility in its natural environment.

2-Another citation for a sentence on lines 180-182 could be added - "Firstly, they need to exit the blood meal, a process that appears to depend on gravity in mosquitoes that agglutinate their blood meal, while it is independent of gravity in those that do not."

Are the text and figures clear and accurate?

Yes, they are clear and accurate. A few minor modifications would add the reader.

1-While references are cited that have measured the elasticity of cell/tissue types, and for PA hydrogels of different compositions, specifically providing these values (along with the reference used for each value) would be helpful. Moreover, expanding Figure 1B to include all PA hydrogels used, and all cell type comparisons would be very helpful. The experiments are robustly done, and making the comparisons between PA hydrogel elasticity and cell type elasticity as clear as possible will strengthen the manuscript and its applicability to the field.

2-The figure legends for soft/medium/stiff hydrogel labels are only in figure 2B, and are the same designation by color throughout figure 2, but they should be applied to all appropriate figures/panels (e.g. Figure 4F-H).

3-Lines 99-102: Authors should define or reframe what they mean by "moved more robustly" as compared to speed.

4-Line 145: Typo "loosing time" should be "losing time"

Do you have suggestions that would help the authors improve the presentation of their data and conclusions?

As noted above, expand Figure 1B to include all PA hydrogel formulations used and cell type comparisons that are noted in the manuscript.

Reviewer #1 (Significance (Required)):

Describe the nature and significance of the advance (e.g. conceptual, technical, clinical) for the field.

Technical:

The authors describe a new technique for assessing movement of the motile stages of Plasmodium using 2D and 3D hydrogels that better mimic the environment that ookinetes and sporozoites encounter in vivo. This approach can be used in future studies to characterize motility defects in transgenic parasite lines or effects of antimalarial drugs on parasite motility.

Biological:

The authors use this approach to characterize ookinetes and sporozoites on differing surfaces, finding that each motile stage uses different strategies for migration, which reflects the different cell types and environments encountered by each stage. Sporozoites may have evolved to not adhere to endothelial cells so that they may be able to identify their target organ, the liver.

Place the work in the context of the existing literature (provide references, where appropriate).

Friedrich Frischknecht has long been involved in studying motile parasite stages and is a leader in this field. The work of his group on this problem is exciting, and it continues to advance our understanding of gliding motility by Plasmodium parasites (and beyond).

Hydrogels of defined porosity and stiffness like those described here have been used for cell culturing for biomechanical and morphological studies, because these hydrogels mimic the natural environment of the cells better than stiff plastic or glass (Caliari and Burdick, Nat Methods 2018 PMID: 27123816).

The motile stages of the parasite experience different local environments during the life cycle, so it makes sense that ookinetes and sporozoites would adapt different strategies to overcome the physical barriers and cell types encountered. In vivo ookinetes were recently imaged to actively move about the blood meal before they traverse the midgut epithelial cells. This approach allowed for in vivo ookinetes to be imaged and the distance traveled/speed of the ookinetes to be measured, but this is a very involved process and is demanding timewise (Trisnadi and Barillas-Murya 2020 mSphere PMID: 32878934). This in vitro hydrogel approach described in Ripp, et al. may mimic what is happening to ookinetes in vivo, and be an easier model system to study ookinete motility. Sporozoites have been imaged at the injection site of the skin of mouse ears, and were found to move quite quickly and freely for a brief time at the site of inoculation, similarly to what Ripp, et al. found in this study in the 3D hydrogels that mimic the skin environment (Hopp, et al. eLife 2015 PMID: 26271010). This further supports the use of these hydrogels as a model system for assessing parasite motility in sporozoites.

State what audience might be interested in and influenced by the reported findings.

Cell biologists in general and parasitologists in particular will be interested in the reported findings.

Define your field of expertise with a few keywords to help the authors contextualize your point of view. Indicate if there are any parts of the paper that you do not have sufficient expertise to evaluate.

Plasmodium transmission

****Referees cross-commenting****

Agreed. Some expansion in descriptions and figure designations will help the reader, as noted in the reviews.

Reviewer #2 (Evidence, reproducibility and clarity (Required)):

****Summary:****

The manuscript details measurements of plasmodium parasite ookinete and sporozoite movement on formulated substrates and how the movement on these substrates pertains to the in vivo condition. Conclusions are then drawn relating to the biological obstacles both the ookinete and sporozoite come across on their respective journeys.

****Major comments:****

The conclusions drawn are reasonable however the authors do not do a good job of explaining to the reader why they chose the substrates they used in their analysis. The lack of this information makes the results extremely difficult to interpret.

The claims are reasonable but they need to be expressed in a way that is understood to the reader. I do not think that further experimentation is necessary but more thorough interpretation of the data needs to take place. The authors claim differences without providing statistics for instance.

****Minor comments:****

I have made numerous comments to the pdf I was able to download and read and will forward this edited file to the journal. Addressing these comments will greatly improve the clarity of the paper.

Reviewer #2 (Significance (Required)):

This work advances our understanding of parasite ookinete and sporozoite motility.

****Referees cross-commenting****

I have re-read my own review and those of the two other reviewers. I feel we are all in agreement that with more attention to detail this manuscript is an interesting and publishable body of work.

Reviewer #3 (Evidence, reproducibility and clarity (Required)):

****Summary****

Transmission of the malaria parasite between vertebrate hosts is dependent on a mosquito vector, and requires the parasite to develop and migrate within the mosquito. Ripp and colleagues describe for the first time the deployment of polyacrylamide (PA) hydrogels to study the motile transmission stages of the rodent malaria parasite *Plasmodium berghei*, the ookinete and the sporozoite.

The ookinete migrates from the blood meal in the mosquito stomach, transforms into an oocyst that produces sporozoites that in turn must travel to the mosquito salivary glands. Upon injection into the skin of the next host, the sporozoite must enter the blood stream to migrate to the liver. Both the ookinete and the sporozoite move by a form of motility termed gliding motility, which is dependent on parasite substrate contact.

The authors used microscopy to acquire images at timed intervals of the moving parasites and measured parasite speed of movement and trajectory using a manual tracking software plugin. Ookinetes were assayed on 2D PA gels, while sporozoites were assayed on 2D and in 3D PA hydrogels, as well as in follow up work with different cell types in vitro. The sporozoite work utilized two mutant *P. berghei* lines with known glass-surface and skin motility phenotypes (a coronin and a heat shock protein 20 knock-out mutant) to confirm that the observed movement in PA hydrogels effectively replicate sporozoite migration in skin. Finally the authors use cytochalasin-D, a known inhibitor of sporozoite motility to show that the PA hydrogel system can also be used to test antibodies and drugs against the parasite sporozoite stage.

The PA hydrogels used in this study offers two unique advantages to established methods. Firstly, they can be used to study motility in 2D as well as 3D. Secondly, the pore size of the PA hydrogels can be precisely altered so to mimic the differences in substrate elasticity the ookinete and sporozoite encounter on their migratory journey.

Using this system Ripp and colleagues show that: 1) Ookinetes remain motile for over 20 hours on hydrogels and that ookinetes adjust their direction of travel from circular on soft, to more linear on stiffer substrate, which is reflected in a greater distance travelled over time on stiffer substrate. 2) By using this system they can construct 3D hydrogels that allow sporozoites to move in a manner akin to that observed by in vivo intradermal imaging. 3) Sporozoites do not adjust the migration path but do adjust their speed depending on (2D) substrate stiffness with a reduced movement observed on soft hydrogels and endothelial cells in vitro.

The model system of the rodent malaria parasite *P. berghei* in combination with the mosquito vector *Anopheles stephensi* a well-established, safe and effective system to produce and study the transmission stages of the malaria parasite.

****Major corrections****

The data is well-presented in a compact paper where the text is well written and accompanied by neat figures. The presentation and readability of the manuscript is greatly enhanced nice use of small and simple descriptive figure panels that directly guides the reader into the experimental set up for each figure. The experiments are appropriately controlled and overall

the conclusions are supported sufficiently by the experiments and the data provided. The manuscript is recommended for publication with some modest revisions. Below I outline recommendations that the authors better annotate some of the figures and revise colour schemes to enhance ease of interpretation, and undertake some further minor explanations of some concepts and the conclusions drawn.

Figure 2. The figures are nice and clean, and labelling of the different substrates in panel 2B is very clear and makes the figure easy to read. The absence of labels for the different substrates, although correlated by colour to figure 2B, makes 2C, 2D, 2E and 2G more difficult to read. Some sort of abbreviated label along X axis (e.g. Soft, Med and Stiff) would make these figures more intuitive to read. This also applies to 2F, however, here the colours annotated in 2B have changed significance to (also) denote different type of motility. Here I would suggest labeling x-axis with substrate elasticity as suggested above but also have a unified colour / pattern scheme for the different categories of motility across substrates to make it more intuitive to compare categories between substrates.

Line 132 / Figure 3. I might have missed this, but could you please explain a bit clearer what conclusion can be drawn from that the coronin mutant is migrating slower than wild type in larger pore but not smallpore PA hydrogel - does it mean that smaller pore size PA gel better mimics skin elasticity?

Conversely, is the overall higher speed observed in the larger pore size hydrogel at all reflecting which pore size mimics skin better? Is this lower speed in smaller pore gels simply due to physical constraint on the parasite or actively regulated by the parasite? You show that the speed in both small and large pore size hydrogel falls within the range of normal speed in skin - does skin itself varies in in elasticity? In the discussion, in the context of regularity of trajectories, you mention that skin is heterogenous in composition does that mean it varies in stiffness? In addition, how does the observed faster motility in softer 3D gels relate to enhanced motility on stiffer / reduced motility on softer 2D substrates? Please comment on these issues, probably best suited to discussion part.

Line 505 / Figure 4C. Cell name acronyms in figure not explained in figure legend, which cell lines are fibroblasts, endothelial cells and hepatocytes. Is the data in 4C from a single experiment - no error bars? In figure legend to Figure 4, in contrast to the other figures, no number of biological repeats is mentioned.

Figure 4F-H. Same as comment regarding Figure 2, difficult to read figure as substrate is not decoded by colour (e.g. by inset legend) nor by annotation of x-axis. Here made even more difficult as one has to remember colour scheme from figure 2 and in same figure (panel 4C) uses the same colour scheme to denote different cell lines.

****Minor corrections****

Line 91 - here it would be beneficial to briefly in once sentence describe microscopy / image acquisition and image analysis method used. Is described only in methods section in brevity. Especially as is main methods used throughout manuscript.

Line 121 / 217 - Explain significance of BSA, why / how activates motility and reference if appropriate.

Line 153 - explain significance of pericytes what is rational behind testing with these (touch upon it on line 156)?

Line 206 - strongly bend, should be bent?

Line 277 - Specify *Anopheles stephensi*

Reviewer #3 (Significance (Required)):

This paper would be of interest to malaria parasite biologists and any parasitologist that studies motile parasites. The impact of this paper is two-fold. Firstly, it presents a versatile and adjustable new improved system to study gliding motility of apicomplexan parasites in 2D as well as 3D. The motility of the transmission stages of malaria parasites have previously been studied in vivo and in vitro. However the new PA hydrogel system presented here uniquely allows fine tuning of 3D pore size and 2D substrate stiffness and thereby for the first time investigate how parasite motility is specifically affected by substrate elasticity since other systems commonly used to study motility such as matrigel is not tunable. Secondly, it uncovers new aspects of biology of the motile transmission stages of the malaria parasite. It shows for the first time that the motile parasite can sense and respond to changes in substrate elasticity by changing pattern and / or speed of movement upon encountering substrates of varying stiffness. The authors hypothesise that this has evolved to allow the ookinete to adapt its motility as it transitions from the soft blood meal to the stiffer midgut epithelium and basal lamina, and similarly allows sporozoites to avoid attachment to softer endothelia as it travels from the bite site through the blood stream to the liver.

Keywords for main reviewer expertise: Malaria, *Plasmodium berghei*, genetic manipulation, host-parasite interactions

****Referees cross-commenting****

Agreed. Subject to the revisions outlined in the review this article is suitable for publication

Authors' Response to Reviewers (transfer from Review Commons)

Reviewer #1 (Evidence, reproducibility and clarity (Required)):

The authors establish a new strategy for assessing the motility of *Plasmodium berghei* ookinetes and sporozoites using 2D and 3D polyacrylamide hydrogels that mimic the environments encountered by each parasite stage in vivo. In vivo imaging experiments are difficult and do not easily allow for changes to experimental parameters, so this in vitro technique is important for characterizing the motility of these parasite stages. For example, the authors characterize the coronin(-) and hsp20(-) parasite lines, whose movement have previously been described in skin and on glass; the hydrogels mimic the elasticity of the skin environment, and so the motility of sporozoites in the hydrogels is a better model for motility than on glass slides. This in vitro strategy that mimics the in vivo environment may be useful in characterizing additional mutant parasite lines or antimalarial drugs that may cause defects in motility at either stage.

A: Thanks for this concise summary of our work and for recognizing the potential in translational research for one of our new assays that allows investigation of *Plasmodium* sporozoite migration in vitro by essentially mimicking the skin phase of infection.

In addition to this methodology, the authors describe the motility of ookinetes and sporozoites and find that each stage uses different strategies for their motility, reflecting the different environments they encounter. The migration path of ookinetes is circular on soft surfaces and linear on stiffer surfaces, reflecting the movement of the ookinete from the soft bloodmeal in the midgut to the less soft epithelial cells of the midgut membrane. Sporozoites are motile for a short period of time, and do not migrate well on soft hydrogel or endothelial cells. The authors suggest the sporozoites are not adhering to the soft endothelial cells of the vasculature on their way to the liver (which is more stiff), and possibly plays a role in the sporozoites finding the correct tissue to invade.

A: Thanks also for this summary of the fundamental concepts that our work uncovered.

Major comments:

Are the key conclusions convincing?

Their results are convincing and the authors describe the data appropriately. The number of cells analyzed for each experiment is adequate and experiments are appropriately controlled.

Should the authors qualify some of their claims as preliminary or speculative, or remove them altogether?

The authors appropriately describe their data, and use suitable language when speculating how the results of the hydrogel models relate to biological events in the life cycle of the motile parasite stages. Claims are appropriately tempered given the data presented.

Would additional experiments be essential to support the claims of the paper?

No additional experiments are needed to support the claims made in this work.

Are the data and the methods presented in such a way that they can be reproduced?

Yes, the materials and methods section describes how experiments were performed, or refers to the appropriate primary literature (Refs 25-28) to describe how data was generated and analyzed. Additional details here or in the main text that directly link an acrylamide/bisacrylamide formulation to specific cell/tissue types would clarify the use of each.

A: We have added these details, also in response to reviewer 2, see line 196-199 and Figure 1B.

Are the experiments adequately replicated and statistical analysis adequate?

Yes, the authors have a large number/adequate number of cells counted for each figure, and use appropriate statistical tests to comment on significance.

Minor comments:

Are prior studies referenced appropriately?

In large part, prior publications are appropriately cited. Some additional citations would be helpful:

1-There could be a citation on line 65, about the in vivo imaging to characterize Plasmodium motility in its natural environment.

A: We added a few references, now line 70,71.

2-Another citation for a sentence on lines 180-182 could be added - "Firstly, they need to exit the blood meal, a process that appears to depend on gravity in mosquitoes that agglutinate their blood meal, while it is independent of gravity in those that do not."

A: We also added two new references here and one cited in the text previously, now line 226.

Are the text and figures clear and accurate?

Yes, they are clear and accurate. A few minor modifications would add the reader.

1-While references are cited that have measured the elasticity of cell/tissue types, and for PA hydrogels of different compositions, specifically providing these values (along with the reference used for each value) would be helpful. Moreover, expanding Figure 1B to include all PA hydrogels used, and all cell type comparisons would be very helpful. The experiments are robustly done, and making the comparisons between PA hydrogel elasticity and cell type elasticity as clear as possible will strengthen the manuscript and its applicability to the field.

A: We have added the numbers in the methods part (lines 367-371) and Figure 1B and also added a statement in the text, which reads "...with the soft hydrogels (5% AA/0.03% BIS) reflecting closest the elasticity of endothelial cells and the intermediate hydrogels (5% AA/0.3% BIS) reflecting the elasticity of dermis and liver" (lines 196-199). We also added a brief discussion to the end of the discussion including also recent work on red blood cells (lines 328-335).

2-The figure legends for soft/medium/stiff hydrogel labels are only in figure 2B, and are the same designation by color throughout figure 2, but they should be applied to all appropriate figures/panels (e.g. Figure 4F-H).

A: We changed the figure accordingly, and agree that it looks now more comprehensible, thanks.

3-Lines 99-102: Authors should define or reframe what they mean by "moved more robustly" as compared to speed.

A: We changed the sentence which now reads "After 20-26 hours of motility, a higher fraction of ookinetes was still motile on soft than on stiff gels but they were fastest on stiff gels", line 113.

4-Line 145: Typo "loosing time" should be "losing time"

A: corrected

Do you have suggestions that would help the authors improve the presentation of their data and conclusions?

As noted above, expand Figure 1B to include all PA hydrogel formulations used and cell type comparisons that are noted in the manuscript.

A: We adapted this as well as figure 2 and 4 as the reviewer suggested.

Reviewer #1 (Significance (Required)):

Describe the nature and significance of the advance (e.g. conceptual, technical, clinical) for the field.

Technical:

The authors describe a new technique for assessing movement of the motile stages of Plasmodium using 2D and 3D hydrogels that better mimic the environment that ookinetes and sporozoites encounter in vivo. This approach can be used in future studies to characterize motility defects in transgenic parasite lines or effects of antimalarial drugs on parasite motility.

Biological:

The authors use this approach to characterize ookinetes and sporozoites on differing surfaces, finding that each motile stage uses different strategies for migration, which reflects the different cell types and environments encountered by each stage. Sporozoites may have evolved to not adhere to endothelial cells so that they may be able to identify their target organ, the liver.

A: Thanks, we could not have put it better.

Place the work in the context of the existing literature (provide references, where appropriate).

Friedrich Frischknecht has long been involved in studying motile parasite stages and is a leader in this field. The work of his group on this problem is exciting, and it continues to advance our understanding of gliding motility by Plasmodium parasites (and beyond).

A: Thanks for the generous and encouraging words.

Hydrogels of defined porosity and stiffness like those described here have been used for cell culturing for biomechanical and morphological studies, because these hydrogels mimic the natural environment of the cells better than stiff plastic or glass (Caliari and Burdick, Nat Methods 2018 PMID: 27123816).

The motile stages of the parasite experience different local environments during the life cycle, so it makes sense that ookinetes and sporozoites would adapt different strategies to overcome the physical barriers and cell types encountered. In vivo ookinetes were recently imaged to actively move about the blood meal before they traverse the midgut epithelial cells. This approach allowed for in vivo ookinetes to be imaged and the distance traveled/speed of the ookinetes to be measured, but this is a very involved process and is demanding timewise (Trisnadi and Barillas-Murya 2020 mSphere PMID: 32878934). This in vitro hydrogel approach described in Ripp, et al. may mimic what is happening to ookinetes in vivo, and be an easier model system to study ookinete motility. Sporozoites have been imaged at the injection site of the skin of mouse ears, and were found to move quite quickly and freely for a brief time at the site of inoculation, similarly to what Ripp, et al. found in this study in the 3D hydrogels that mimic the skin environment (Hopp, et al. eLife 2015 PMID: 26271010). This further supports the use of these hydrogels as a model system for assessing parasite motility in sporozoites.

A: Thanks again for this correct statement of the advances that we hope our work provides for both fundamental and translational research.

State what audience might be interested in and influenced by the reported findings.

Cell biologists in general and parasitologists in particular will be interested in the reported findings.

A: We see it as our challenge working on a complex parasite to provide insights that are of interest to the general cell biology audience and hope that we occasionally achieve this, e.g. see Douglas et al. PLoS Biol 2018 and Spreng et al., EMBO J 2019 for work that illuminates the divergent biology of actin filaments and microtubules, respectively. Here, we hope that also the biophysically minded cell and developmental will be intrigued by the way Plasmodium adapts to the different environments that it encounters at different developmental stages. As for parasitologists, they appreciate more and more the importance of physical parameters as a driving force for parasites, a topic now enshrined in the research program physics of parasitism funded by the German Research Foundation.

Define your field of expertise with a few keywords to help the authors contextualize your point of view. Indicate if there are any parts of the paper that you do not have sufficient expertise to evaluate.

Plasmodium transmission

Referees cross-commenting

Agreed. Some expansion in descriptions and figure designations will help the reader, as noted in the reviews.

Reviewer #2 (Evidence, reproducibility and clarity (Required)):

Summary:

The manuscript details measurements of plasmodium parasite ookinete and sporozoite movement on formulated substrates and how the movement on these substrates pertains to the in vivo condition. Conclusions are then drawn relating to the biological obstacles both the ookinete and sporozoite come across on their respective journeys.

Major comments:

The conclusions drawn are reasonable however the authors do not do a good job of explaining to the reader why they chose the substrates they used in their analysis. The lack of this information makes the results extremely difficult to interpret.

The claims are reasonable but they need to be expressed in a way that is understood to the reader. I do not think that further experimentation is necessary but more thorough interpretation of the data needs to take place. The authors claim differences without providing statistics for instance.

A: Thanks for these critical comments and the many helpful suggestions on the pdf file, which we took to heart and used as starting points to add much more in depth descriptions. This, in our view, made the manuscript indeed easier to understand and read without expanding the text too much. We really appreciate the reviewers effort and time. Please see the file with the marked changes to see the extensive additions we made, they are too numerous to list here.

****Minor comments:****

I have made numerous comments to the pdf I was able to download and read and will forward this edited file to the journal. Addressing these comments will greatly improve the clarity of the paper.

A: Thanks. As mentioned above, we expanded on the vast majority of the really helpful suggestions, which we think now provides a better (i.e. more readable) manuscript.

Reviewer #2 (Significance (Required)):

This work advances our understanding of parasite ookinete and sporozoite motility.

****Referees cross-commenting****

I have re-read my own review and those of the two other reviewers. I feel we are all in agreement that with more attention to detail this manuscript is an interesting and publishable body of work.

Reviewer #3 (Evidence, reproducibility and clarity (Required)):

****Summary****

Transmission of the malaria parasite between vertebrate hosts is dependent on a mosquito vector, and requires the parasite to develop and migrate within the mosquito. Ripp and colleagues describe for the first time the deployment of polyacrylamide (PA) hydrogels to study the motile transmission stages of the rodent malaria parasite *Plasmodium berghei*, the ookinete and the sporozoite.

The ookinete migrates from the blood meal in the mosquito stomach, transforms into an oocyst that produces sporozoites that in turn must travel to the mosquito salivary glands. Upon injection into the skin of the next host, the sporozoite must enter the blood stream to migrate to the liver. Both the ookinete and the sporozoite move by a form of motility termed gliding motility, which is dependent on parasite substrate contact.

The authors used microscopy to acquire images at timed intervals of the moving parasites and measured parasite speed of movement and trajectory using a manual tracking software plugin. Ookinetes were assayed on 2D PA gels, while sporozoites were assayed on 2D and in 3D PA hydrogels, as well as in follow up work with different cell types in vitro. The sporozoite work utilized two mutant *P. berghei* lines with known glass-surface and skin motility phenotypes (a coronin and a heat shock protein 20 knock-out mutant) to confirm that the observed movement in PA hydrogels

effectively replicate sporozoite migration in skin. Finally the authors use cytochalasin-D, a known inhibitor of sporozoite motility to show that the PA hydrogel system can also be used to test antibodies and drugs against the parasite sporozoite stage.

The PA hydrogels used in this study offers two unique advantages to established methods. Firstly, they can be used to study motility in 2D as well as 3D. Secondly, the pore size of the PA hydrogels can be precisely altered so to mimic the differences in substrate elasticity the ookinete and sporozoite encounter on their migratory journey.

Using this system Ripp and colleagues show that: 1) Ookinetes remain motile for over 20 hours on hydrogels and that ookinetes adjust their direction of travel from circular on soft, to more linear on stiffer substrate, which is reflected in a greater distance travelled over time on stiffer substrate. 2) By using this system they can construct 3D hydrogels that allow sporozoites to move in a manner akin to that observed by in vivo intradermal imaging. 3) Sporozoites do not adjust the migration path but do adjust their speed depending on (2D) substrate stiffness with a reduced movement observed on soft hydrogels and endothelial cells in vitro.

The model system of the rodent malaria parasite *P. berghei* in combination with the mosquito vector *Anopheles stephensi* a well-established, safe and effective system to produce and study the transmission stages of the malaria parasite.

A: Thanks for this correct summary of our work and for understanding the value of our study for both fundamental biological and translational research.

****Major corrections****

The data is well-presented in a compact paper where the text is well written and accompanied by neat figures. The presentation and readability of the manuscript is greatly enhanced nice use of small and simple descriptive figure panels that directly guides the reader into the experimental set up for each figure. The experiments are appropriately controlled and overall the conclusions are supported sufficiently by the experiments and the data provided. The manuscript is recommended for publication with some modest revisions. Below I outline recommendations that the authors better annotate some of the figures and revise colour schemes to enhance ease of interpretation, and undertake some further minor explanations of some concepts and the conclusions drawn.

A: Thanks also to reviewer 3 for the constructive critique, that along with those of the other reviewers helped us to improve our paper.

Figure 2. The figures are nice and clean, and labelling of the different substrates in panel 2B is very clear and makes the figure easy to read. The absence of labels for the different substrates, although correlated by colour to figure 2B, makes 2C, 2D, 2E and 2G more difficult to read. Some sort of abbreviated label along X axis (e.g. Soft, Med and Stiff) would make these figures more intuitive to read. This also applies to 2F, however, here the colours annotated in 2B have changed significance to (also) denote different type of motility. Here I would suggest labeling x-axis with substrate elasticity as suggested above but also have a unified colour / pattern scheme for the different categories of motility across substrates to make it more intuitive to compare

categories between substrates.

A: We changed the figure as suggested by the reviewer, which in our view made them easier to understand. Thank you for the suggestion.

Line 132 / Figure 3. I might have missed this, but could you please explain a bit clearer what conclusion can be drawn from that the coronin mutant is migrating slower than wild type in larger pore but not smallpore PA hydrogel - does it mean that smaller pore size PA gel better mimics skin elasticity?

A: We think that the skin has a mix set of “pores” corresponding to both the large and small pores in our gels, which of course are also not of uniform distribution. Hence we think that the mixed distribution of parasites migrating in both gels reflects best the distribution of migration in the skin. *Coronin(-)* parasite have a defect in adhesion (Bane et al. Plos Path 2016) and possible force production (unpublished work). So likely in larger pores they are less good gliders as they fail to find enough substrate to adhere to. We expanded on this in the text, see lines 155-158, by adding a sentence: “This might reflect the decreased capacity of *coronin(-)* sporozoites to attach to surfaces (34), which could lead to impaired gliding in gels with large pores (i.e. less available substrate surface), but is compensated by gels with small pore (i.e. more available substrate surface).”

Conversely, is the overall higher speed observed in the larger pore size hydrogel at all reflecting which pore size mimics skin better? Is this lower speed in smaller pore gels simply due to physical constraint on the parasite or actively regulated by the parasite?

A: We think that the skin provides a wide range of “pore sizes” as the gels do. Sporozoites indeed show very different behavior in the skin, not all move, some move very robustly, some get “stuck” for some time before continuing to move. The lower speed could be due to both. Clearly physical constrain must be one reason, however we can't exclude that the parasite does sense the environment and respond. This will be an interesting question to answer.

You show that the speed in both small and large pore size hydrogel falls within the range of normal speed in skin - does skin itself varies in in elasticity? In the discussion, in the context of regularity of trajectories, you mention that skin is heterogenous in composition does that mean it varies in stiffness? In addition, how does the observed faster motility in softer 3D gels relate to enhanced motility on stiffer / reduced motility on softer 2D substrates? Please comment on these issues, probably best suited to discussion part.

A: very good questions, we now address them in more detail in the discussion, see lines 259-270. “Interestingly, we found sporozoites moving within soft PA hydrogels, which feature narrow pores (30), indicating that substrate softness does not impair motility in a 3D setting. Sporozoites slowed down in hydrogels with decreasing pore size, likely due to steric hindrance imposed by the polymeric network. However, we cannot exclude that sporozoites also sense and respond to the substrate and modulate their behavior. This could be addressed by investigating sporozoites with e.g. calcium sensitive dyes as they migrate through the gels, as previous work showed that calcium

is elevated in sporozoites migrating on flat surfaces in vitro (42). Also, motility clearly depends on a rapid turnover (i.e. formation and disruption) of adhesion sites (36). Sporozoites forming very tight adhesions are slower than those forming weaker ones, while those forming very weak adhesions cannot move. Hence, sporozoites with a slight defect in adhesion might not move on a flat and stiff substrate anymore as seen for coronin(-) sporozoites (34) but are still moving fine in a 3D environment.“

We hope this makes our thoughts and interpretations as well as the open questions that our work poses more accessible. One interesting follow-up question would be if sporozoites in “old” skin move different to “young” skin, assuming that old skin is less elastic.

Line 505 / Figure 4C. Cell name acronyms in figure not explained in figure legend, which cell lines are fibroblasts, endothelial cells and hepatocytes. Is the data in 4C from a single experiment - no error bars? In figure legend to Figure 4, in contrast to the other figures, no number of biological repeats is mentioned.

A: This part of the paper started out with a grant proposal that intended to find the surface proteins on endothelial cells necessary for sporozoite gliding. We performed many experiments first on HUVEC then on other vascular cells and always failed to see sporozoite movement. To the point where we got really desperate until we started to understand that sporozoites might just not move on these types of cells. We eventually performed experiments on different lines but as single experiments and quantitatively analyzed them. We now state this in the text clearly but removed the cell lines where we performed imaging experiments only once in Figure 4C. Please note we performed at least three experiments on all the cell lines but just never made any movies as nothing moved. Despite the non-significance of the test for data in Figure 4C, we believe the data to be sufficient as shown and described in the text. Additionally, we now show new data supporting our observations that were obtained independently by colleagues in Paris, who are now also listed as co-authors and did three independent experiments (new Figure 4A).

Figure 4F-H. Same as comment regarding Figure 2, difficult to read figure as substrate is not decoded by colour (e.g. by inset legend) nor by annotation of x-axis. Here made even more difficult as one has to remember colour scheme from figure 2 and in same figure (panel 4C) uses the same colour scheme to denote different cell lines.

A: Thanks, we changed this accordingly.

****Minor corrections****

Line 91 - here it would be beneficial to briefly in once sentence describe microscopy / image acquisition and image analysis method used. Is described only in methods section in brevity. Especially as is main methods used throughout manuscript.

A: We added “...performed time-lapse live cell imaging using a wide field fluorescence microscope for either fluorescent, phase- or differential interference contrast (DIC) imaging”, now line 102.

Line 121 / 217 - Explain significance of BSA, why / how activates motility and

reference if appropriate.

A: We added “Sporozoite motility is induced by bovine serum albumin (BSA) (33)”, now line 141.

Line 153 - explain significance of pericytes what is rational behind testing with these (touch upon it on line 156)?

A: We added “another type of vascular cell”, now line 179. We also explained in more detail how we first struggled to get sporozoites move on different vascular cells, just literally trying as many as we could get our hands on.

Line 206 - strongly bend, should be bent?

A: Agreed and changed accordingly.

Line 277 - Specify *Anopheles stephensi*

A: Done, now line 351, reads “which are the Indian mosquito species usually used in laboratory experiments with *P. berghei*”.

Reviewer #3 (Significance (Required)):

This paper would be of interest to malaria parasite biologists and any parasitologist that studies motile parasites. The impact of this paper is two-fold. Firstly, it presents a versatile and adjustable new improved system to study gliding motility of apicomplexan parasites in 2D as well as 3D. The motility of the transmission stages of malaria parasites have previously been studied in vivo and in vitro. However the new PA hydrogel system presented here uniquely allows fine tuning of 3D pore size and 2D substrate stiffness and thereby for the first time investigate how parasite motility is specifically affected by substrate elasticity since other systems commonly used to study motility such as matrigel is not tunable. Secondly, it uncovers new aspects of biology of the motile transmission stages of the malaria parasite. It shows for the first time that the motile parasite can sense and respond to changes in substrate elasticity by changing pattern and / or speed of movement upon encountering substrates of varying stiffness. The authors hypothesise that this has evolved to allow the ookinete to adapt its motility as it transitions from the soft blood meal to the stiffer midgut epithelium and basal lamina, and similarly allows sporozoites to avoid attachment to softer endothelia as it travels from the bite site through the blood stream to the liver.

A: Thanks for this nice summary stating the importance of our work for the understanding of the malaria life cycle. As already stated for reviewer 1, we also hope that our work will be of interest to general cell biologists studying motile cells using biophysical approaches.

Keywords for main reviewer expertise: Malaria, *Plasmodium berghei*, genetic manipulation, host-parasite interactions

Referees cross-commenting

Agreed. Subject to the revisions outlined in the review this article is suitable for publication

15th Jan 2021

Dear Prof. Frischknecht,

Thank you for the submission of your revised manuscript to EMBO Molecular Medicine. I am pleased to inform you that we will be able to accept your manuscript pending the following final amendments:

- 1) With the beginning of the new year, we encountered high number of submissions, so that our data editors were not able to process all received manuscripts. Therefore, we will send you the document with data editor's suggestions as soon as our data editors process your manuscript. Please do not submit your revised manuscript before we send you the file with data editor's suggestions. Thank you for your understanding.
- 2) Manuscript Type: You submitted your article as a report and in our opinion, this is an appropriate format, therefore please reduce the number of figures to max. 3. Please check "Author Guidelines" for more information.
<https://www.embopress.org/page/journal/17574684/authorguide#reportarticleguide>
- 3) Figures: Please upload individual, high-resolution figure files. For more information on figure presentation please check "Author Guidelines".
<https://www.embopress.org/page/journal/17574684/authorguide#datapresentationformat>
- 4) Movies: Rename movie files to "Movie EV1" etc. (also in the text) and zipp their legends with respective movie file.

The authors performed the requested editorial changes.

29th Jan 2021

Dear Prof. Frischknecht,

We are pleased to inform you that your manuscript is accepted for publication and is now being sent to our publisher to be included in the next available issue of EMBO Molecular Medicine.

Please read below for additional IMPORTANT information regarding your article, its publication and the production process.

Congratulations on your interesting work,

Zeljko Durdevic

Follow us on Twitter @EmboMolMed
Sign up for eTOCs at embopress.org/alertsfeeds

Corresponding Author Name: Friedrich Frischknecht

Manuscript Number: EMM-2021-13933